# Transient Characteristics of Fluidic Pintle Nozzle in a Solid Rocket Motor

Dongfeng Yan [1],*, Zehang Zhao [1], Anchen Song [2], Fengming Li [1], Lu Ye [1], Ganchao Zhao [1] and Shan Ma [1]

1   Flight Technology College, Civil Aviation Flight University of China, Guanghan 618307, China;
    zzh001118@foxmail.com (Z.Z.); fmlnuaa@126.com (F.L.); yelucafuc@163.com (L.Y.);
    13658152702@163.com (G.Z.); nirvaa.shanma@gmail.com (S.M.)
2   Beijing System Design Institute of the Electro-Mechanic Engineering, Beijing 100039, China;
    songanchen1991@foxmail.com
*   Correspondence: yandongfeng@cafuc.edu.cn; Tel.: +86-191-8108-6997

**Abstract:** The fluidic pintle nozzle, a new method to control the thrust of a solid rocket motor, has been proposed in recent years by combining the pintle with the aerodynamic throat (fluidic throat). The study of static characteristics has proved that it has a remarkable effect on thrust control. To study the transient characteristics of the fluidic pintle nozzle, 2D transient simulations of a fluidic pintle nozzle propulsion system were conducted, employing dynamic meshing techniques. The Reynolds-averaged Navier–Stokes equations were meticulously solved, implementing a k–$\omega$ SST turbulence model. The thrust control principle of the fluid pintle nozzle was studied, and the wave structure was summarized. The transient characteristics of the secondary flow opening, secondary flow closing, pintle moving forward (pressure rise), and pintle moving backward (pressure drop) were obtained, and the effects of the injection angle and injection port position were studied. The response process after injection can be roughly divided into three stages: pressure propagation, pressure oscillation, and equilibrium stability, with time distributions of 0.4%, 5.39%, and 94.21%, respectively. In the process of the pintle moving forward, the rate of combustion chamber pressure increases and thrust decreases gradually because of the arc wall of the nozzle throat upstream, and the process of throats moving backward is just the opposite. Compared with the condition with a maximum throat opening and no secondary flow, the thrust of the condition with a minimum throat opening and a 0.3-flow-ratio secondary flow is increased by 80.95%. Under conditions of constrained flow ratio, the injection angle of the secondary flow ostensibly exerts negligible influence on the dynamic modulation of thrust. Nevertheless, it remains evident that a reduction in throat opening accentuates the impact of reverse injection. Furthermore, the proximity of the injection port to the head of the pintle is directly proportional to the efficacy of thrust control.

**Keywords:** solid rocket motor; thrust control; pintle motor; aerodynamic throat; fluidic throat; transient characteristics





## 1. Introduction

Solid rocket motors are characterized by their simplicity, high reliability, ease of maintenance, compact size, rapid response, and ease of storage and transportation [1]. They have been widely applied in various fields, including tactical and strategic missiles, rocket boosters, altitude and trajectory control motors, ejection systems, and more. In terms of production and operability, solid rocket motors possess irreplaceable advantages and play a crucial role. In the current and future landscape of information warfare, as weapon systems advance in technologies such as interception and breakthrough confrontation, there is an urgent need for intelligent thrust-adjustable solid rocket motor technology [2]. Intelligent thrust control technology, encompassing both thrust magnitude and direction, allows for the rational allocation of propellant energy in missile motors, meeting the requirements of various operational tasks. Missiles equipped with thrust-adjustable capabilities will



possess higher levels of intelligence, maneuverability, versatility, range coverage, and operational adaptability. Realizing stochastic control of thrust would denote a momentous advancement in solid rocket motor technology.

Adjusting the thrust of solid rocket motors is not easily achievable, as once the solid propellant ignites, it is challenging to extinguish. Feasible thrust control methods for solid rocket motors currently include multiple extinguishing/igniting cycles, pre-designing solid propellant columns that meet mission requirements, changing the burning rate of the propellant through special means, and changing the nozzle throat area. Among these, changing the nozzle throat area is relatively easy to implement and falls into two main categories: mechanical and fluidic methods.

A typical mechanical type is the pintle motor [3,4], which controls the thrust by changing the effective gas flow area (the throat formed by the nozzle wall and pintle) through the movement of the pintle. Research on the pintle motor began in 1968, and the large motor (267 kg of propellant mass) achieved a thrust control of 567–3900 kg [5], but ignition and sustained combustion were not smooth. Subsequent research on pintle motors has continued to progress, and in recent years, significant achievements have been made. From 2013 to 2017, H.-G. Sung [6,7] conducted a transient numerical simulation study on reciprocating the pintle using dynamics mesh, and obtained certain results on the pressure and thrust and their sensitivity. Song [8] used RNG k-ε to verify the results of the cold flow experiment and showed that the burning rate response characteristics of solid propellant have an important impact on the operation of the pintle. Dong-Sung Ha [9] conducted a cold flow experiment and thermal test on the pintle motor, and obtained the schlieren picture of expansion state at the outlet, confirming that this control method has a highly compensative effect and is expected to be used in space–space integrated equipment.

Typical methods of flow types are vortex valves [10] and a fluidic nozzle throat (FNT). A vortex valve stands as a fluidic control element devoid of mechanical components. It modulates thrust by imparting a tangential fluid spray into the motor, inducing rotational motion within the combustion gas. This rotational effect amplifies the impedance of the primary flow, subsequently modifying the pressure within the combustion chamber and regulating the rate of combustion gas generation. Wei et al. conducted Particle Image Velocimetry (PIV) experiments and simulations on the vortex valve in 2018, and obtained the influence of swirl characteristics and structural parameters [11]. The concept of fluidic throat (aerodynamic throat) first appeared in 1957 [12], where the introduction of secondary flow was used to compress the primary flow, leading to changes in the flow area of the primary flow. This mechanism is employed to control thrust. In 2010, Ali [13] first applied this concept to solid rocket motors, and due to the unique burning rate characteristics of solid propellants, it proved to be highly effective. In 2017, Guo [14] conducted a detailed study on the fluidic throat of a solid rocket motor through a cold flow experiment, hot experiment, and numerical simulation, which fully verified the effectiveness of this method in a solid rocket motor. There are also studies on the use of secondary flow to reduce motor noise [15]. In essence, secondary flow technology belongs to the category of jet or jet in crossflow. The technology of flight control by secondary flow has been widely used in recent years [3,16,17].

The Fluidic Pintle Nozzle (FPN) is a hybrid approach combining mechanical and fluidic methods. The FPN is designed to introduce a secondary flow at the head of the pintle, so that the primary flow area can be more effectively regulated by combining the mechanical control of the pintle with fluidic control [18]. Meanwhile, the relatively low temperature secondary flow can cover the surface of the pintle, which weakens the ablation of the pintle by the primary flow at high temperature and pressure. In addition, solid propellant can be filled into the hollow pintle, which is expected to alleviate the problem of excessive demand for an air source in the secondary flow.

The pintle itself in the motor, however, will cause a complex wave structure, flow separation, and vortex backflow [6,7], while the solid propellent has the pressure-burning rate response characteristic, which makes the pintle motor have a typical unstable characteristic

during operation. The secondary flow not only makes the transient characteristics of the flow field in the motor more complicated, but also interferes with the strong unsteady characteristics of the flow field during the opening and closing of the secondary flow [14]. This may affect the flow field, which may cause excessive local load and environmental noise of the aircraft, and induce the chattering of the aircraft structure. Although some studies have shown that this method can effectively control thrust, it is limited to static studies. The transient characteristics of FPN are not clear at present, so it is necessary to study systematically.

This paper, utilizing dynamic mesh, conducts a two-dimensional numerical simulation to study the transient characteristics of FPN. The research delves into transient processes such as secondary flow opening and closure and forward and backward movement of the pintle, with a focus on the variations in combustion chamber pressure and thrust. Furthermore, the impact of injection angles and nozzle port positions on dynamic characteristics is discussed.

## 2. Principle of FPN

General solid rocket motor propellants are pre-designed: for the same propellant, its density, characteristic velocity, burning rate coefficient, pressure index, and density are all fixed. The thrust coefficient is only related to the geometric structure and has little variation range. Therefore, the most direct way to change the thrust is to change the throat area and the propellant burning area. Pre-machined specific propellant shapes can also achieve changes in the burning area, but can no longer be controlled once ignited.

The thrust control principles of FPN are very similar to those of the fluidic throat, as described in reference [18]. Here, the working principles of FPN are elucidated in conjunction with Figure 1.

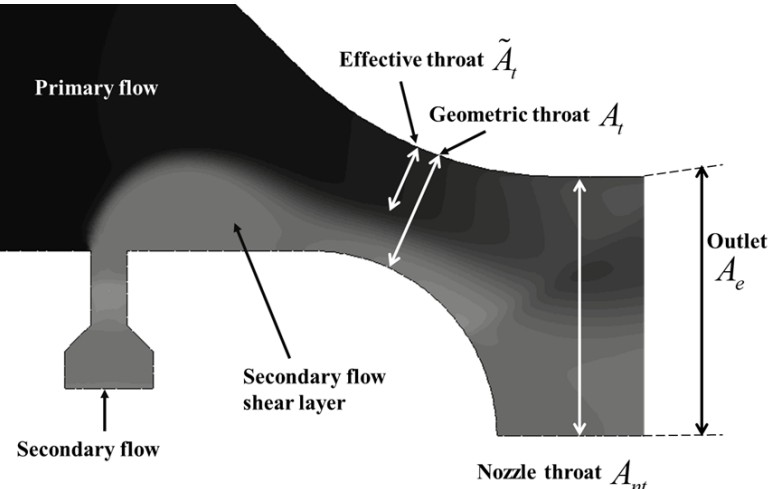

**Figure 1.** Schematic diagram of FPN.

Within a pintle motor, a geometric throat materializes between the pintle wall and the nozzle wall. In instances where the geometric throat area $A_t$ surpasses that of the nozzle throat area $A_{nt}$, signifying that the pintle has not advanced to a specific position, the motor throat retains the characteristics of the nozzle throat. The outlet area $A_e$ remains unaltered. Following the injection of secondary flow, the secondary flow compresses the spatial domain of the primary flow, giving rise to a novel primary flow profile. This profile emerges through the extrusion of two fluid streams, embodying a flexible configuration wherein the effective throat area $\tilde{A}_t$ is as shown inf in Figure 1.

In addition, in order to eliminate the influence of the working medium and temperature, the modified flow parameters and flow ratio are introduced:

$$\dot{w} = \dot{m}\sqrt{T/\mu} \tag{1}$$

$$f_w = \frac{\dot{w}_s}{\dot{w}_o} = \frac{\dot{m}_s \sqrt{T_s \mu_o}}{\dot{m}_o \sqrt{T_o \mu_s}} \qquad (2)$$

where $\mu$ is the molar mass of the gaseous working fluid, $T$ is the temperature of the medium, $\dot{m}$ is flow rate, the subscript $o$ represents the primary flow, and $s$ represents the secondary flow.

After defining the modified flow ratio, the effect of secondary flow temperature and molar mass on the result can be normalized, which has a certain generalization advantage. Guo [14] has verified in his research the effect of the modified flow ratio on predicting the secondary flow and primary flow different temperatures and molar masses.

In this study, the thrust in the numerical results is calculated according to equation $F = \dot{m}v_e + A_e(P_e - P_a)$, where $P_e$ is the pressure of motor outlet and $P_a$ is the pressure of the external environment. This equation takes the volume formed by the inner wall and the outlet section of the nozzle as the control body to integrate, which is not affected by the propellant type and the internal geometric mechanism of the motor, and can be calculated only by obtaining the exit interface parameters.

## 3. Models and Numerical Methods

### 3.1. Geometric Models and Boundary Conditions

The maximum geometric throat of the selected motor is the nozzle throat. When the area formed by the pintle and the nozzle wall is equal to the nozzle throat, the pintle position is defined as the pintle opening, which is 100%. The minimum geometric throat is defined as the pintle opening being at 0% when the pintle extends into nozzle throat. The size of the structure and the movement range of the pintle are shown in Figure 2, and the parameters are shown in Table 1. The volume of the cavity from the primary flow inlet to the throat is $3.2 \times 10^4$ mm$^3$.

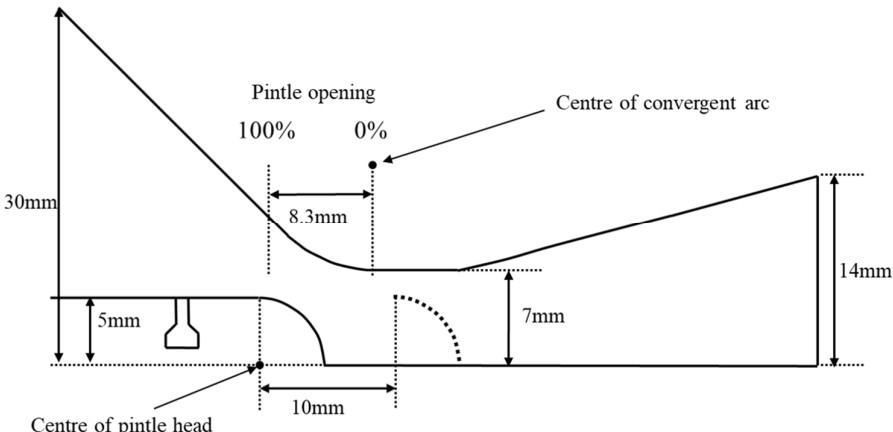

**Figure 2.** Geometric model and range of motion of the pintle.

**Table 1.** Geometric model parameters.

| Component | Value |
| --- | --- |
| Combustion chamber diameter | 60 mm |
| Nozzle throat diameter | 14 mm |
| Pintle diameter | 10 mm |
| Nozzle outlet diameter | 28 mm |
| Convergent half angle | 45° |
| Half-angle expansion | 15° |
| Expansion ratio | 4 |
| Free volume of the cavity | $3.2 \times 10^4$ mm$^3$ |

In order to more effectively simulate transient characteristics, the primary inlet is configured as a mass flow inlet with combustion rate feedback. Recognizing that an excessive flow from secondary injection can result in a substantial load on the gas supply system, the modified flow ratio is set to 0.3. Consequently, the primary flow rate under the condition of 100% pintle opening without secondary injection is employed as the benchmark for the modified flow ratio (0.2 kg/s). The outlet is configured as a pressure outlet, utilizing standard atmospheric parameters at sea level. With 100% opening and no secondary injection, the pressure in the combustion chamber is 2.03 MPa, and the outlet pressure is set at 0.086 MPa (0.85 atm). The boundary conditions and configurations for dynamic mesh settings are delineated in Figure 3. The combustion gas parameters are as indicated in Table 2.

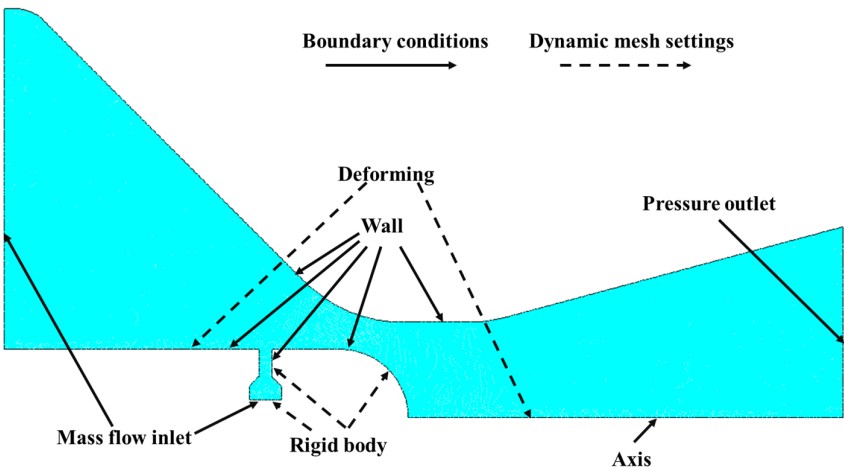

**Figure 3.** Boundary conditions and configurations for dynamic mesh settings.

**Table 2.** Combustion gas parameters.

| Component | Parameter | Value |
|---|---|---|
| Propellant | Burning rate coefficient | 6.7 |
| | Pressure index | 0.24 |
| | Density | 1700 kg/m$^3$ |
| | Burning area | $1.53 \times 10^{-2}$ mm$^2$ |
| Gas | Molar mass | 26.3157 g/mol |
| | Specific heat | 1.63 kJ/(kg·K) |
| | Thermal conductivity | 0.285 w/(m·k) |
| | Temperature of primary flow | 3000 K |
| | Temperature of secondary flow | 1789 K |
| | Mass flow of secondary flow | 0.07968 kg/s |

In the simulation of the pintle opening and closing process, the time step is set to $10^{-7}$ ms. In the simulation of the pintle motion, the time step is set to $10^{-6}$ ms, and the total duration is 66.6 ms. The pintle's range of motion spans from 100% to 0% opening, with additional clearance both before and after. The displacement is 10 mm, and both forward and backward movements occur at a speed of 150 mm/s, as illustrated in Figure 2. After the cessation of the pintle motion, the calculation continues until the combustion chamber pressure stabilizes. At 4.67 ms, the geometric throat area ($A_t$) equals the nozzle throat area ($A_{nt}$), and after 66.67 ms, the geometric throat area no longer undergoes variations. Figure 4 illustrates the relationship between time and the actual geometric throat area ($A_t$) during the pintle advancing process.

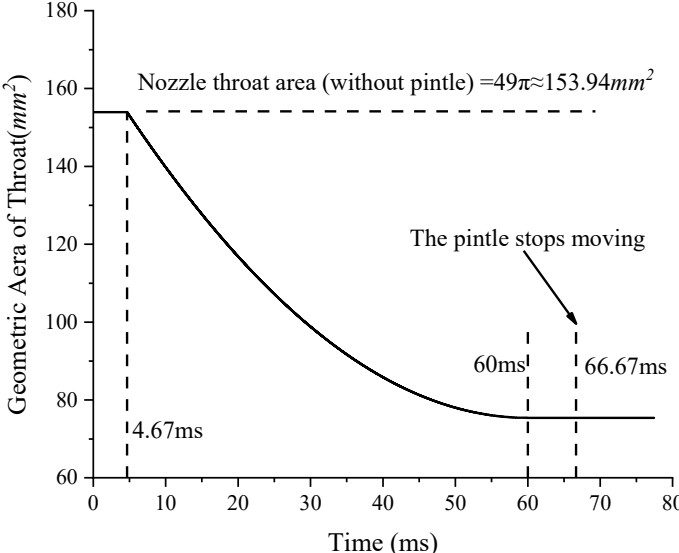

**Figure 4.** Relationship between time and geometric throat area during the pintle advancing process.

Figure 5 illustrates a schematic diagram of the secondary injection angle and the injection port positions, $\alpha$ represents the injection angle, with $\alpha$ being equal to 90° indicating a vertical injection, and angles greater than 90° indicating a reverse injection. $R_p$ denotes the radius of the pintle, while $L_i$ designates the location of the injection port, defined as the distance from the head of the pintle to the injection port. The dimensionless parameter $L_i/R_p$ serves to characterize the relative position of the injection port.

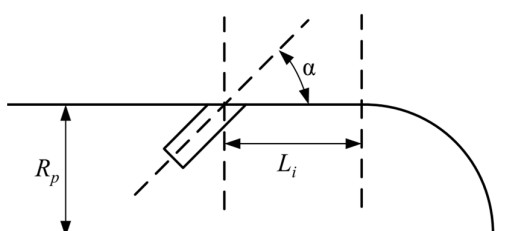

**Figure 5.** Schematic diagram of the secondary flow injection position.

Within the pintle motor, the convergence of the pintle wall and the converging section wall of the nozzle gives rise to a novel geometric throat. The new geometric throat is essentially a circumferential gap of a conical sector ring. The geometric throat area can be determined based on the surface area formula for a cone, yielding:

$$A_t = S = \pi \frac{AE}{AD}(DB^2 - DC^2) \tag{3}$$

Considering the unique pintle configuration and diverse design requirements, the numerical model for the aforementioned calculation method is illustrated in Figure 6 and can be expressed as follows:

$$A_t = S = \pi \frac{y_b}{DB}(DB^2 - DC^2) \tag{4}$$

Taking the minimum value, the geometric throat area is:

$$A_t = S = Min\left[\pi \frac{y_b}{DB}(DB^2 - DC^2)\right] \tag{5}$$

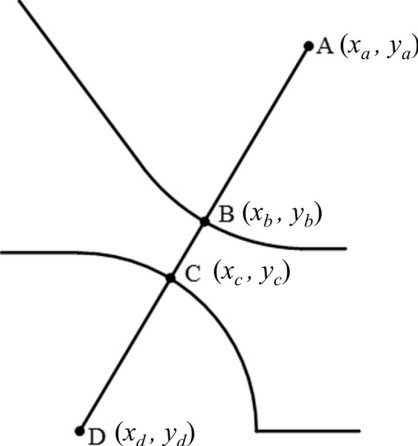

**Figure 6.** Calculation of the geometric throat area in a pintle motor.

*3.2. Numerical Methods*

Complex physical and chemical processes occur in the combustion chamber and nozzles of a solid rocket motor during secondary flow injection. To simplify the calculations, the following assumptions are introduced into the numerical model in this paper:

(1) Monophasic flow, excluding the consideration of solid particulates introduced into the propellant;
(2) The gas is treated as an ideal gas, adhering to the equations of state for an ideal gas;
(3) No consideration for radiative heat transfer, neglect of gravity, and absence of volume forces;
(4) Adiabatic boundaries, devoid of thermal exchange between the external environment and the entire flow field.

This study primarily employs the k–ω SST model based on the Reynolds-averaged Navier–Stokes (RANS) equation.

In Reynolds-averaging, the instantaneous solution of the Navier–Stokes equation is decomposed into mean components (time-averaged or ensemble-averaged) and fluctuating components. For velocity components:

$$u_i = \overline{u}_i + u_i' \tag{6}$$

where $\overline{u}$ represents the mean component, and $u_i'$ represents the fluctuating component. Similarly, the same decomposition is applied to pressure, energy, and species.

The expression in the above form is used to substitute the flow variable into the instantaneous continuity equation and the momentum equation, and the time average or ensemble average is taken. The ensemble average momentum equation can be expressed in the form of the Cartesian tensor as follows:

$$\frac{\partial \rho}{\partial t} + \frac{\partial}{\partial x_i}(\rho u_i) = 0 \tag{7}$$

$$\frac{\partial}{\partial t}(\rho u_i) + \frac{\partial}{\partial x_j}(\rho u_i u_j) = -\frac{\partial p}{\partial x_i} + \frac{\partial}{\partial x_j}\left[\mu\left(\frac{\partial u_i}{\partial x_j} + \frac{\partial u_j}{\partial x_i} - \frac{2}{3}\delta_{ij}\frac{\partial u_l}{\partial x_l}\right)\right] + \frac{\partial}{\partial x_j}\left(-\rho\overline{u_i'u_j'}\right) \tag{8}$$

where $u_i$ represents the Reynolds mean velocity component without the mean sign omitted, $u_i'$ represents the fluctuating velocity, $\mu$ represents the dynamic viscosity, $\rho$ represents the density, $p$ represents the pressure, and $\delta_{ij}$ represents Kronecker's delta.

This paper, considering both computational accuracy and cost, adopts the shear-stress transport k–ω SST model. The k–ω SST model, proposed by Menter [19], utilizes the k–ω model near the wall and the k–ω model in the far-field, free-flow region. By combining the advantages of both models, the k–ω SST model offers a balance between accuracy and

robust performance. This model is particularly suitable for calculating pressure gradient flows, considering cross-diffusion terms, making it applicable near and far from the wall. Previous research indicates that this model performs well in computations involving high Reynolds numbers, transonic, and supersonic flows [20–22]. The conservation equations for this model are:

$$\frac{\partial}{\partial t}(\rho k) + \frac{\partial}{\partial x_i}(\rho k u_i) = \frac{\partial}{\partial x_j}\left(\Gamma_k \frac{\partial k}{\partial x_j}\right) + G_k - Y_k + S_k \tag{9}$$

$$\frac{\partial}{\partial t}(\rho \omega) + \frac{\partial}{\partial x_j}(\rho \omega u_j) = \frac{\partial}{\partial x_j}\left(\Gamma_\omega \frac{\partial \omega}{\partial x_j}\right) + G_\omega - Y_\omega + D_\omega + S_\omega \tag{10}$$

where $k$ represents turbulent kinetic energy and $\omega$ is the specific rate of dissipation. $G_k$ represents the generation term for turbulent kinetic energy (induced by mean velocity gradients), $G_\omega$ is the generation term for the specific rate of dissipation $\omega$, and $\Gamma_k$ and $\Gamma_\omega$ are the effective diffusion coefficients for $k$ and $\omega$, respectively. $Y_k$ and $Y_\omega$ represent the turbulent dissipation due to turbulence production for $k$ and $\omega$, respectively. $S_k$ and $S_\omega$ are user-defined source terms, and $D_\omega$ represents the cross-diffusion term.

The k–ω SST model is built upon the foundation of the standard k–ω and standard k–ω models. To bridge the two, a cross-diffusion term $D_\omega$ is introduced. For the specific equations, please refer to the literature [19]. Definitions for other parameters can be found in the ANSYS Fluent Theory Guide published by Ansys Inc. (Canonsburg, PA, USA) [23].

The simulations were conducted utilizing a 2D axisymmetric, pressure-based, double precision, implicit, unsteady solver within Fluent®. The numerical method is grounded in the finite volume method. Gradient interpolation is predicated on the least squares cell method, accompanied by standard pressure interpolation. The density, momentum, and energy equations are instantiated in the first-order upwind format. The pressure–velocity coupling is facilitated through the implementation of the SIMPLEC algorithm. The transient term in time is discretized using a first-order implicit scheme.

### 3.3. Dynamic Mesh

The study employed the smooth spring method and the remeshing method for a mesh update. The details are not reiterated here and can be referred to in the ANSYS Fluent Theory Guide [23]. The specific parameter settings are as follows.

The Spring/Laplace/Boundary layer method is employed in smoothing. The Spring Constant Factor is set to 0.7, and the Laplace Node Relaxation is set to 0.7. Local Cell, Local Face, and Region Face are utilized in Remeshing. The Minimum Length Scale is 0.001 mm, and the Maximum Length Scale is 0.17 mm. The Maximum Cell Skewness is limited to 0.3.

Mesh displays for the "moving forward, $\alpha = 90°$, and $L_i/R_p = 1$" case at t = 0 ms and t = 30 ms are presented in Figure 7.

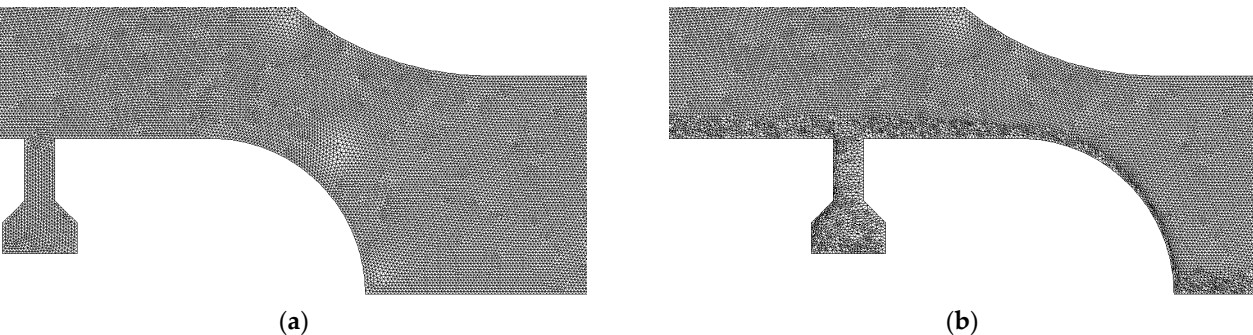

(**a**)  (**b**)

**Figure 7.** Mesh at different pintle positions. (**a**) T = 0.0 ms. (**b**) T = 30 ms.

### 3.4. Calculation Verification

To validate the numerical calculation methodology, the Grid Convergence Index (GCI), as proposed by Roache [24], is utilized for verification. Specifically, the case involving "$f_w$ = 0.3, pintle moves forward, angle = 90° and without injection" has undergone scrutiny. The mesh scales employed encompass 97,134 for the coarse mesh, 135,402 for the medium mesh, and 163,244 for the fine mesh.

As the mesh undergoes refinement, the efficacy of the methodology improves progressively. The impact of further refinement is reduced by the transition from the medium mesh to the fine mesh. The maximum value of GCI12 (the convergence index between fine and medium mesh) is 0.025, a magnitude deemed acceptable within the purview of engineering calculations, as depicted in Figure 8. All calculations within this manuscript are predicated upon the employment of the fine mesh.

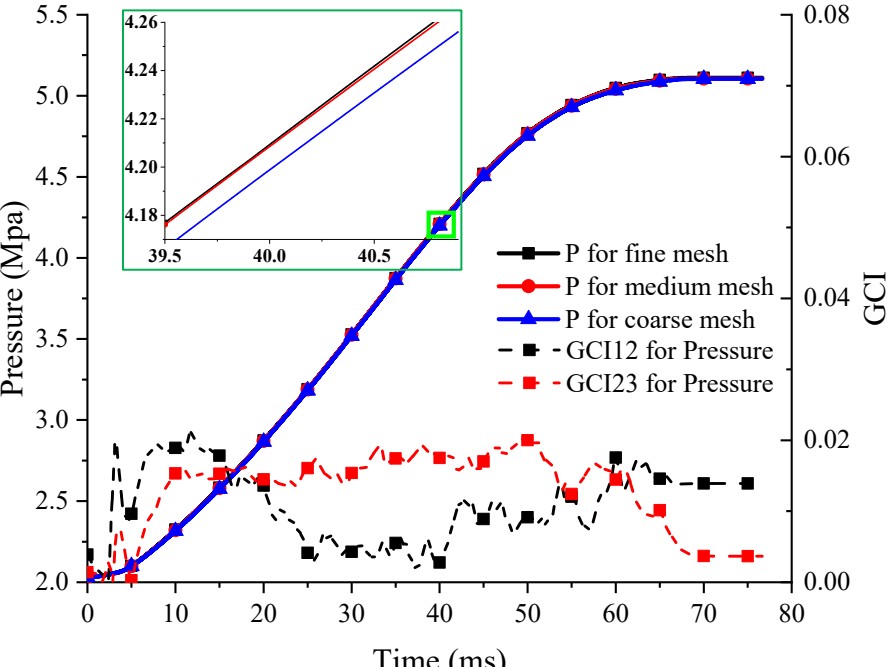

**Figure 8.** Mesh independence verification.

Additionally, pressure data from the cold flow experiment on the pintle motor conducted by Song [8] were selected for validation. The results are depicted in Figure 9. In the figure, "Exp-original" is experimental data, and "CFD-original" is simulation data, both of which are from Song's paper. "CFD-Validation" is the result calculated by authors based on Song's model and data in order to validate the numerical method. The computational outcomes align fundamentally with Song's Computational Fluid Dynamics (CFD) simulations and the empirical findings from cold flow tests. The experimental minimum pressure is 6.35917 MPa, and the maximum is 7.56858 MPa. In the validation calculation, the minimum pressure is 6.3103361 MPa, and the maximum is 7.7046 MPa. The error for the minimum pressure is 0.7%, and for the maximum pressure, it is 1.8%.

Therefore, the simulation could be deemed reasonable.

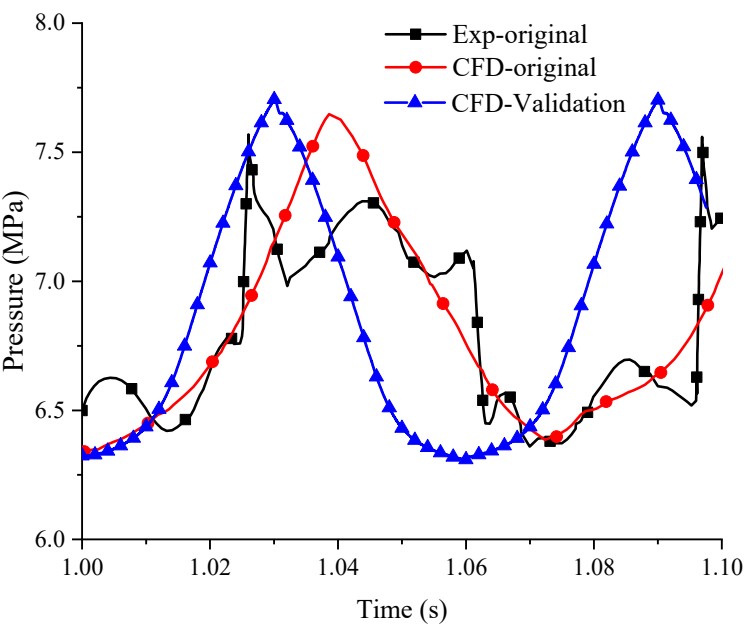

**Figure 9.** Validation compared with experiment result.

## 4. Results and Discussion

### 4.1. Transient Process of Secondary Flow Opening and Closing

#### 4.1.1. Opening

Figure 10 depicts the flow, temperature, and Mach number distributions in the motor after injection, with pintle opening $O_p$ = 0%, injection angle $\alpha$ = 90°, and modified flow ratio $f_w$ = 0.3. Figure 11 illustrates the variation in combustion chamber pressure and wall pressure distribution during the opening process. From the visual representations, the post-secondary injection response process can be broadly categorized into three phases, namely the pressure propagation stage, pressure oscillation stage, and equilibrium stability stage.

Preceding the injection of the secondary flow, the combustion chamber pressure stands at 5.12 MPa. When the high-pressure secondary flow starts injecting, it enters the combustion chamber through the injection ports on the pintle, compressing the primary flow and forming a compression wave (0.02 ms). The primary flow is obstructed, and the compression wave travels upstream until it reflects back to the entire combustion chamber, eventually dissipating, as shown in Figure 11a. The lower-temperature secondary flow is compressed and constrained by the high-pressure primary flow, flowing downstream along the pintle wall. During this stage, the overall combustion chamber pressure remains relatively constant. This phase is referred to as the pressure propagation stage (Figure 10, approximately 0 ≤ t ≤ 0.023 ms; Figure 11b, $t_0$~$t_1$). Subsequently, the secondary flow flows downstream while adhering to the pintle, forming a secondary shear layer. Due to the compression by secondary flow, the effective throat area decreases, leading to an increase in combustion chamber pressure. The momentary high pressure causes the secondary shear layer to thin, increasing the effective throat area and reducing pressure. The transient low pressure thickens the secondary shear layer, decreasing the effective throat area. This is a repetitive process, and although the overall combustion chamber pressure is increasing, the thickness of the secondary shear layer and the combustion chamber pressure exhibit oscillatory characteristics until the combustion chamber pressure no longer shows a decreasing trend. Throughout this process, the thicknesses of the secondary shear layer and pressure oscillate. As the lower-temperature secondary flow gradually fills the vortex region at the head of the pintle, the overall temperature in this area begins to decrease, while the area of the recirculation zone does not change significantly. This stage is referred to as the pressure oscillation stage (Figure 10, approximately 0.023 ≤ t ≤ 0.33 ms; Figure 11b, $t_1$~$t_2$). The primary flow and secondary flow gradually begin to reach equilibrium, and the rate of increase in the combustion chamber pressure gradually decreases until it stabilizes.

The flow also tends to stabilize during this stage, known as the equilibrium stability stage (Figure 10, approximately $0.33 \leq t \leq 5.7$ ms; Figure 11b, approximately $t_2 \sim 5.7$ ms). Research suggests that the response process of injection is related to the free volume [25].

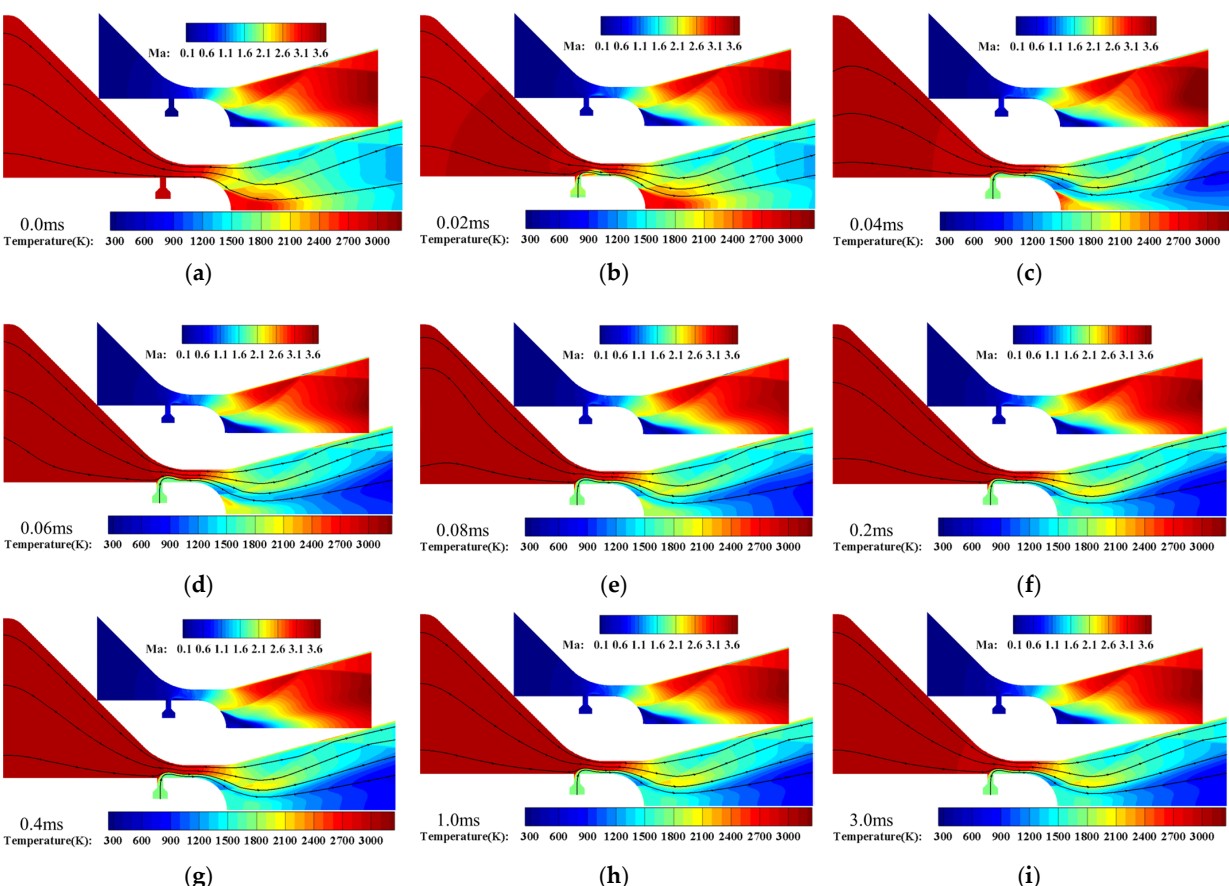

**Figure 10.** Streamlines, temperature, and Mach number contours after injection, opening = 0%, $\alpha = 90°$, $f_w = 0.3$. (**a**) t = 0.0 ms. (**b**) t = 0.02 ms. (**c**) t = 0.04 ms. (**d**) t = 0.06 ms. (**e**) t = 0.08 ms. (**f**) t = 0.2 ms. (**g**) t = 0.4 ms. (**h**) t = 1.0 ms. (**i**) t = 3.0 ms.

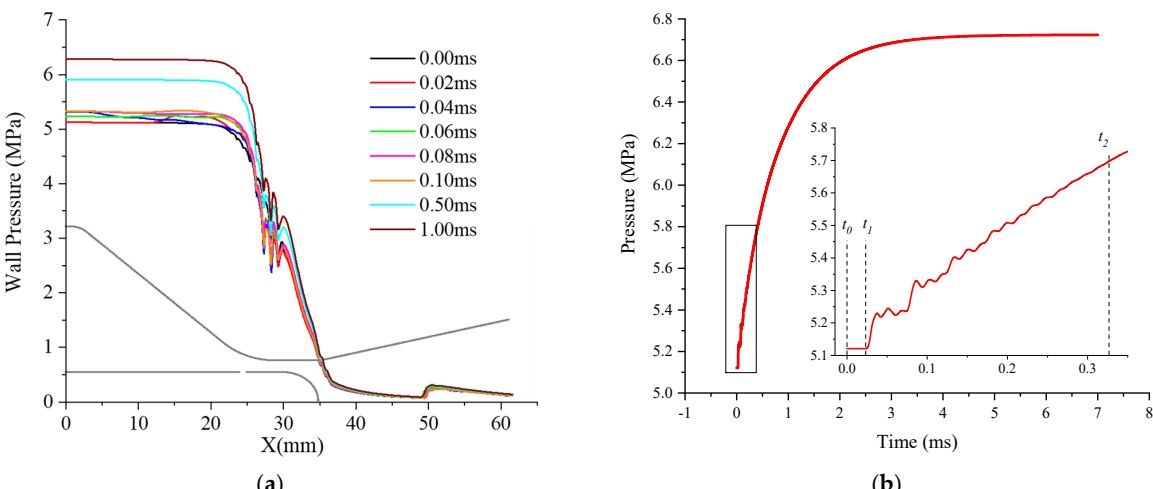

**Figure 11.** Pressure variation after injection, opening = 0%, $\alpha = 90°$, $f_w = 0.3$. (**a**) Pressure distribution along the nozzle wall. (**b**) Combustion chamber pressure.

The injection of secondary flow at the pintle's head results in the compression of the primary flow, giving rise to the formation of a secondary shear layer adhering to the

pintle wall. As depicted in Figures 10 and 12, the secondary flow delineates a noteworthy flow separation vortex region (V2) downstream of the injection port. The secondary flow undergoes compression by the primary flow and subsequently reattaches to the pintle wall, with the point of reattachment designated as A1. Subsequently, the secondary flow attaches to the pintle and completely envelops the head of the pintle. The encapsulation of the low-temperature secondary flow serves to diminish the wall temperature of the pintle, imparting a degree of thermal protection to the component. The gas flow in the curved section of the pintle experiences compression due to the recirculation zone V1, leading to flow separation and the generation of oblique shock waves. However, the location of flow separation is downstream-shifted compared to the case without secondary flow, leading to an upstream shift in the reflected shock position R1. The recirculation zone V1 is then filled by the secondary flow closer to the axis, resulting in higher density and pressure in this region compared to the case without the secondary flow [18]. There exists no distinct boundary of significance between the primary flow and the secondary flow, except with regard to temperature. This is because the injection port is upstream of the throat, and after compression and expansion through the throat, the pressure and Mach numbers of the two flows become similar.

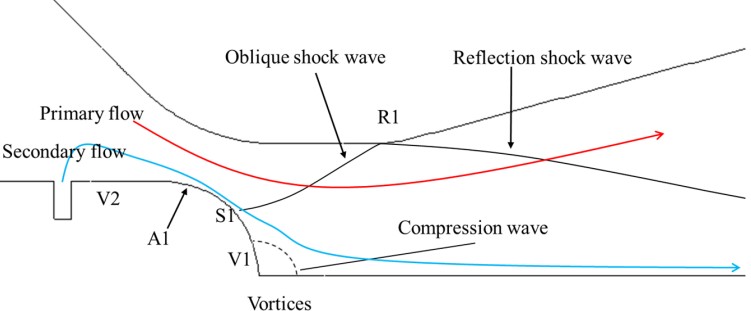

**Figure 12.** Schematic of the flow field wave system.

### 4.1.2. Closing

The process of closing the secondary flow is relatively straightforward. Figure 13 illustrates the combustion chamber pressure variation and wall pressure distribution during the closing process with the pintle opening Op = 0%, injection angle $\alpha = 90°$, and modified flow ratio $f_w = 0.3$.

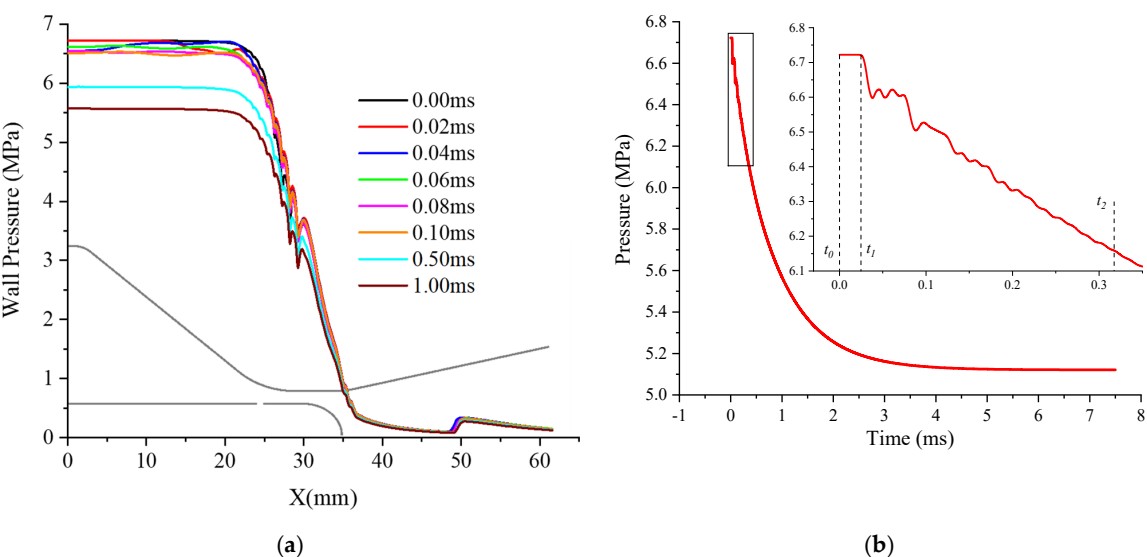

**Figure 13.** Pressure variation and wall pressure distribution after closing the secondary flow, opening = 0%, $\alpha = 90°$, $f_w = 0.3$. (**a**) Pressure distribution along the nozzle wall. (**b**) Combustion chamber pressure.

After closing the secondary flow, the combustion chamber pressure experiences a brief delay of 0.025 ms before starting to oscillate and decline. It continues to steadily decrease starting from 0.32 ms. The recirculation zone downstream of the pintle head gradually starts to be replaced by the primary flow, resulting in a gradual increase in gas temperature. The pressure oscillations during the closing process are evidently smaller than during the opening process, and the time required to reach equilibrium is also similar, approximately 6 ms.

### 4.2. Coupling of Pintle Movement and Secondary Flow

The most distinctive feature of the pintle motor is the thrust control achieved through pintle movement. This section analyzes the transient characteristics during the motion of FPN.

#### 4.2.1. Forward Movement of the Pintle (Pressure Increase Process)

As the pintle progresses downstream, diminishing the geometric throat area, the combustion chamber pressure ascends, thereby culminating in an augmentation of motor thrust. This process is referred to as the forward movement of the pintle, and in some studies, it is also termed the pressure increase process. In FPN, the forward movement of the pintle also causes the downstream displacement of the secondary flow injection port, and the position of the injection port is continuously changing. To compare the impact of secondary flow on the pintle, the forward movement process of a general pintle motor without injection is also provided. Figure 14 shows the temperature and Mach number contours during the forward movement process of the pintle without injection. Figure 15 displays the temperature and Mach number contours during the forward movement process of FPN with an equivalent flow ratio $f_w = 0.3$. Figure 16 delineates the evolution of pressure and thrust across the entire progression.

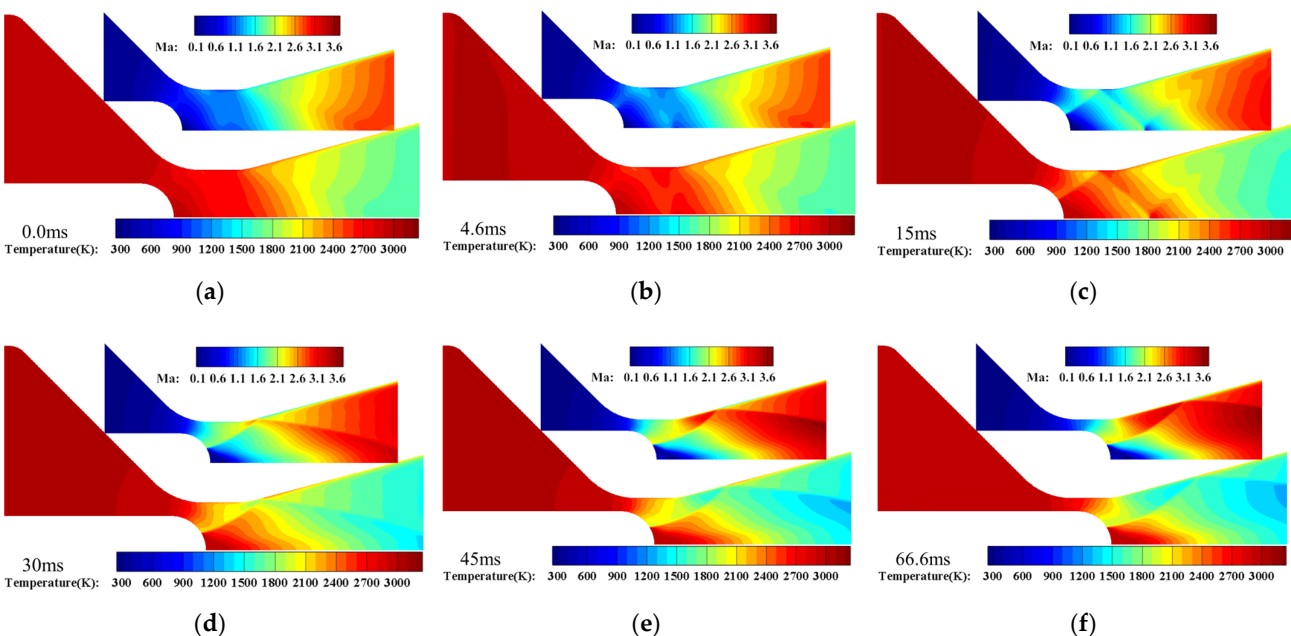

**Figure 14.** Temperature and Mach number contours during the forward movement process of the pintle motor, $\alpha = 90°$, $f_w = 0$. (**a**) t = 0.0 ms. (**b**) t = 4.6 ms. (**c**) t = 15 ms. (**d**) t = 30 ms. (**e**) t = 45 ms. (**f**) t = 66.6 ms.

In the study, the initial flow field is in a steady state. At t = 0 ms, the pintle starts moving. During the forward movement process, when t < 4.6 ms, the pintle has not yet reached 100% opening, meaning that the actual throat at this time is still the nozzle throat. At t = 4.6 ms, the pintle reaches 100% opening, and at this point, the actual throat of

the motor shifts to the annular gap formed between the pintle and the nozzle wall. At t = 60 ms, the throat area formed by the pintle and the nozzle wall reaches its minimum. Subsequently, the pintle continues its downstream movement, maintaining a constant geometric throat area until t = 66.6 ms, at which point the pintle halts its motion, and the combustion chamber pressure reaches a state of stability.

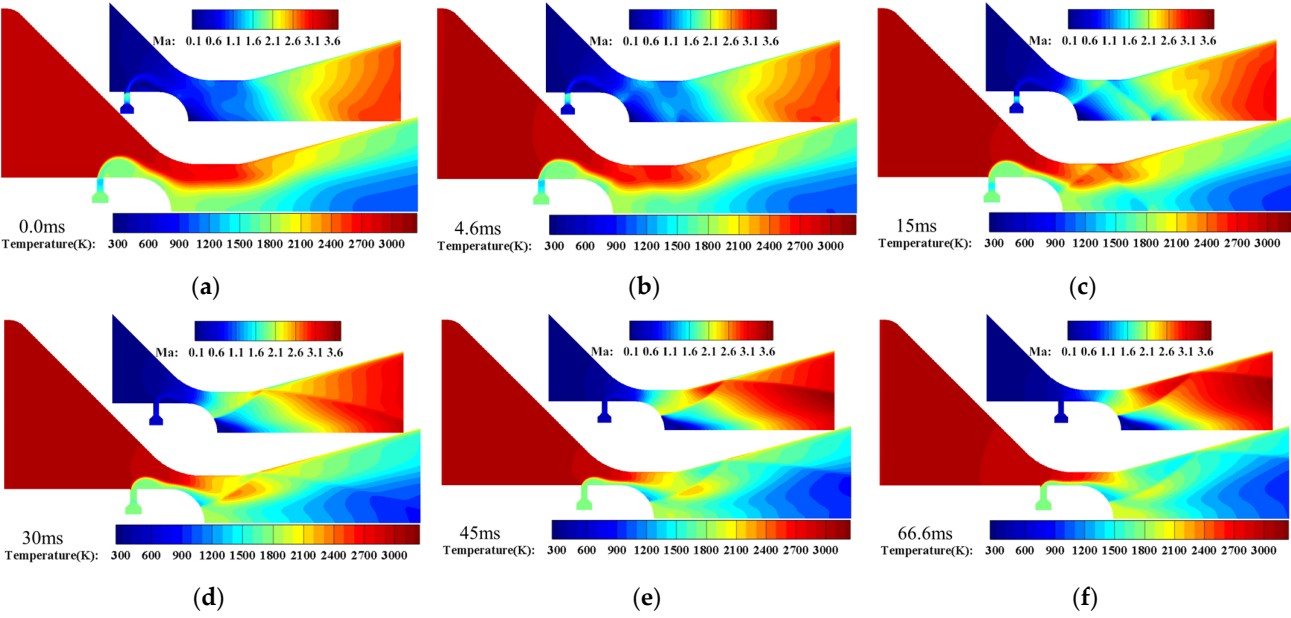

**Figure 15.** Temperature and Mach number contours during the forward movement process of FPN, $\alpha = 90^\circ$, $f_w = 0.3$. (**a**) t = 0.0 ms. (**b**) t = 4.6 ms. (**c**) t = 15 ms. (**d**) t = 30 ms. (**e**) t = 45 ms. (**f**) t = 66.6 ms.

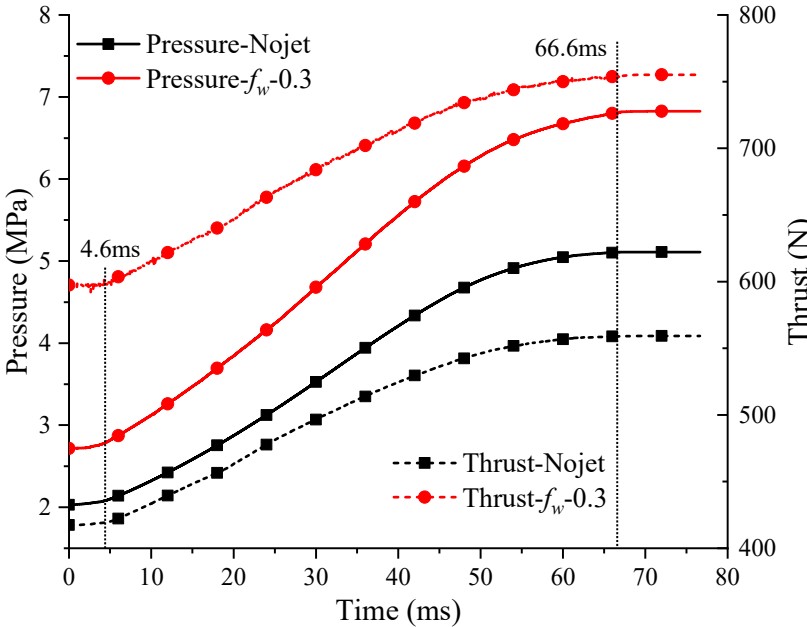

**Figure 16.** Variation of combustion chamber pressure and thrust during the forward movement process.

Comparing the general pintle motor without injection and FPN with secondary flow injection, at t = 0 ms, the sonic surfaces of both motors are located at the nozzle throat. At t = 4.6 ms, when the pintle moves forward to 100% opening, the sonic surfaces of both motors have shifted to the annular gap formed by the pintle and the nozzle. From 0 to 4.6 ms, there is no change in combustion chamber pressure for both motors, as shown in Figure 16. From 4.6 to 60 ms, the pintle continuously moves downstream, and the

position of the injection port also moves accordingly. This sequential motion engenders a gradual upswing in combustion chamber pressure. According to the thrust regulation of FPN, a larger modified flow ratio results in a faster ascent rate. This phenomenon arises due to the heightened flow ratio amplifying the throttling impact of the secondary flow on the primary flow. This, in turn, diminishes the flow area of the primary flow, consequently expediting a more rapid ascent in pressure. It is noteworthy that, during the forward movement as the pintle approaches the throat, the rate of pressure augmentation diminishes. This is attributed to the construction of the upstream arc of the nozzle throat, which gradually slows down the rate of reduction in the geometric throat area, as shown in Figure 4. At t = 60 ms, the geometric throat area reaches its minimum and remains constant thereafter. Thus, during the 60–66.6 ms, the combustion chamber pressure remains essentially unchanged. At t = 66.6 ms, the pintle stops moving, and within the next 5 ms, the pressure gradually stabilizes. In the process of moving forward, the combustion chamber pressure of the general pintle motor increases from 2.03 MPa to 5.11 MPa, and the thrust increases from 417.35 N to 559.27 N. For the FPN, the combustion chamber pressure increases from 2.72 MPa to 6.83 MPa, and the thrust increases from 597.4 N to 755.18 N. Compared with "opening = 100%, $f_w$ = 0", and "opening = 0%, $f_w$ = 0.3", the thrust is increased by 80.95% (from 417.35 N to 755.18 N).

### 4.2.2. Backward Movement of the Pintle (Pressure Decrease Process)

The backward movement of the pintle (pintle moving upstream in the nozzle) increases the geometric throat area, reduces the combustion chamber pressure, and decreases thrust. This process is also referred to as the pressure decrease process in some studies. Figure 17 illustrates the temperature and Mach number distribution during the backward movement of a general pintle motor, Figure 18 depicts the temperature and Mach number distribution for $f_w$ = 0.3 in FPN during the backward movement, and Figure 18 illustrates the changes in pressure and thrust.

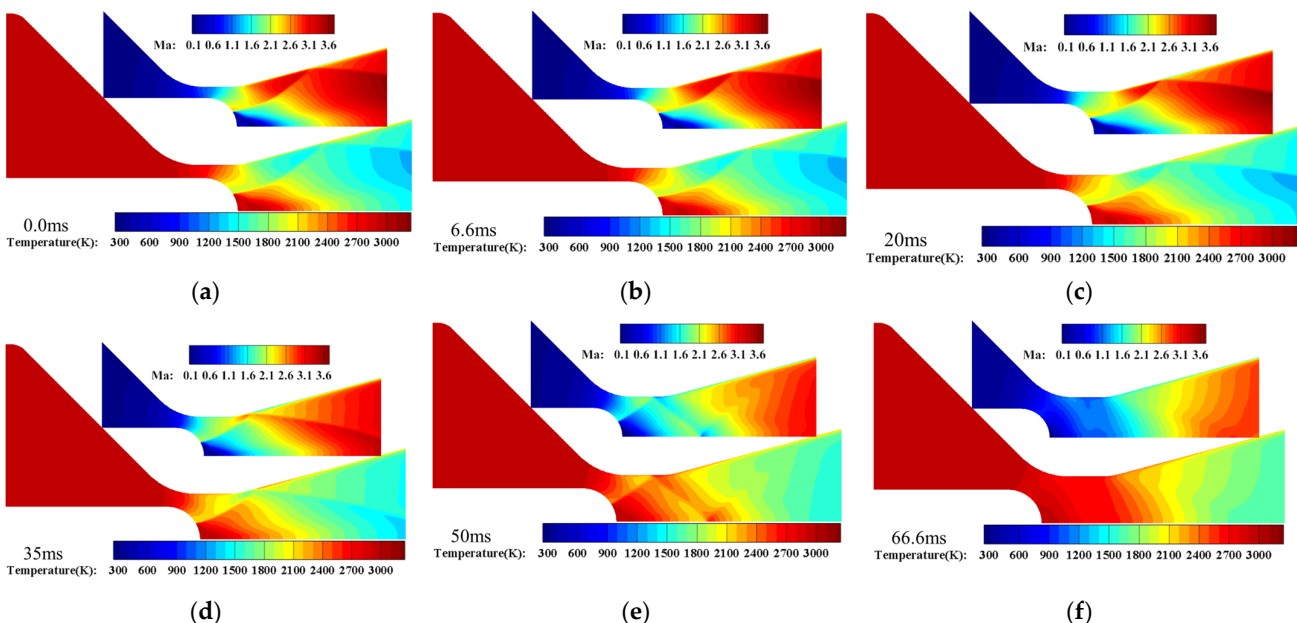

**Figure 17.** Temperature and Mach number contours during the backward movement process of pintle motor, $\alpha$ = 90°, $f_w$ = 0. (**a**) t = 0.0 ms. (**b**) t = 6.6 ms. (**c**) t = 20 ms. (**d**) t = 35 ms. (**e**) t = 50 ms. (**f**) t = 66.6 ms.

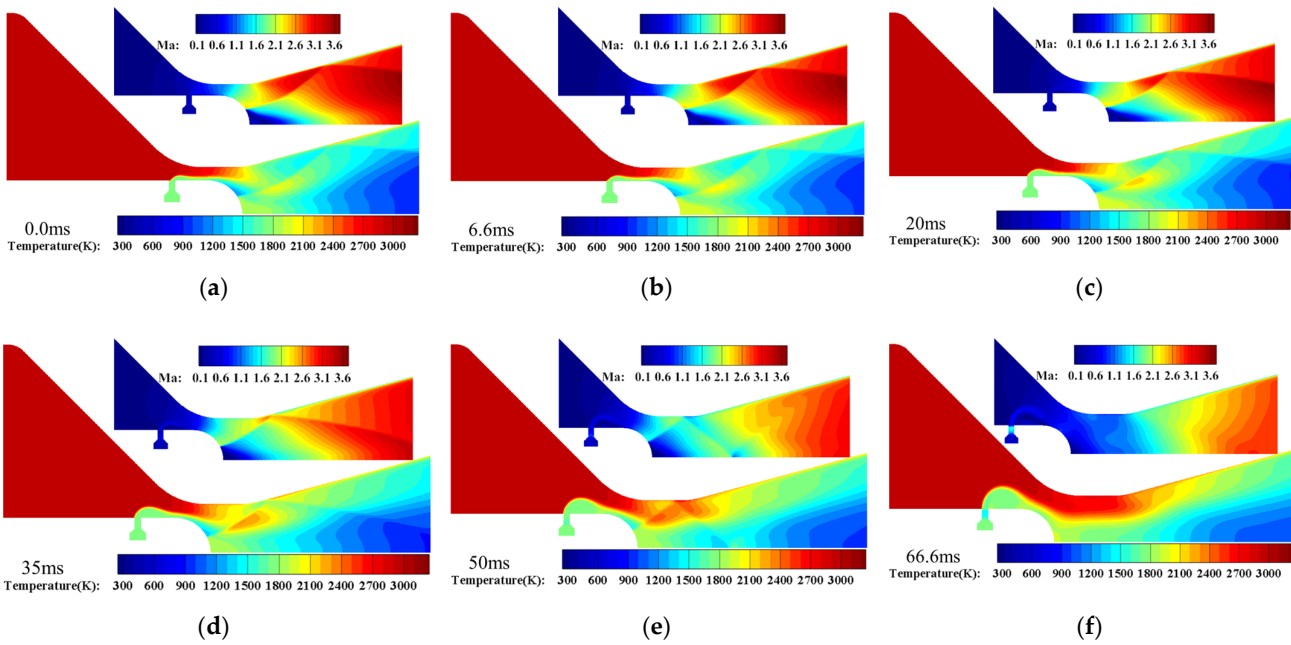

**Figure 18.** Temperature and Mach number contours during the backward movement process of FPN, $\alpha = 90°$, $f_w = 0.3$. (**a**) t = 0.0 ms. (**b**) t = 6.6 ms. (**c**) t = 20 ms. (**d**) t = 35 ms. (**e**) t = 50 ms. (**f**) t = 66.6 ms.

In the investigation of the pintle retraction process, the initial flow field is a steady-state flow field. At t = 0 ms, the pintle initiates movement, and within the time interval $0 < t \leq 6.6$ ms, the geometric throat area remains constant. At t = 6.6 ms, the geometric throat area begins to decrease, and the sonic surface moves upstream. By t = 62 ms, the geometric throat area reaches its maximum value, with the pintle opening is 100%, and the sonic surface starts to shift toward the nozzle throat. At t = 66.6 ms, the pintle ceases its motion and combustion chamber pressure attains equilibrium.

At t = 0 ms, both the general pintle motor and FPN have their sonic surfaces located at the annular gap formed by the pintle and the nozzle wall. Due to the influence of the low-temperature secondary flow, the temperature in the downstream recirculation region of the pintle head within the FPN is noticeably lower. Additionally, the constriction effect of the secondary flow results in a higher combustion chamber pressure. In comparison to Figure 19, during the time from 0 to 6.6 ms, although there is a change in the pintle position, the geometric throat area remains unchanged. This circumstance results in no alteration in combustion chamber pressure and thrust. At t = 6.6 ms, the geometric throat area begins to undergo changes, gradually increasing in size. In the time interval from 6.6 to 62 ms, the geometric throat area continuously increases, leading to a reduction in combustion chamber pressure. Simultaneously, due to the influence of the upstream arc of the nozzle throat, the rate of pressure decrease diminishes. At t = 62 ms, the pintle reaches 100% opening, and the geometric throat area equals the nozzle throat area, initiating the transfer of the sonic surface to the nozzle throat. During this phase, the rate of combustion chamber pressure decrease noticeably slows down. An analysis attributes this to the change in the sonic surface causing a decrease in pressure differential along the pintle wall, leading to a sudden loss of this component of thrust. Approximately 3 ms after the pintle ceases its motion, the combustion chamber pressure stabilizes. In the process of moving backward, the combustion chamber pressure of the general pintle motor decreases from 5.11 MPa to 2.01 MPa, and the thrust decreases from 559.37 N to 416.37 N. For the FPN, the combustion chamber pressure decreases from 6.81 MPa to 2.73 MPa, and the thrust decreases from 755.23 N to 598.5 N. Compared with "opening = 0%, $f_w = 0.3$" and "opening = 100%, $f_w = 0$", the thrust is decreased by 44.87% (from 775.23 N to 416.37 N).

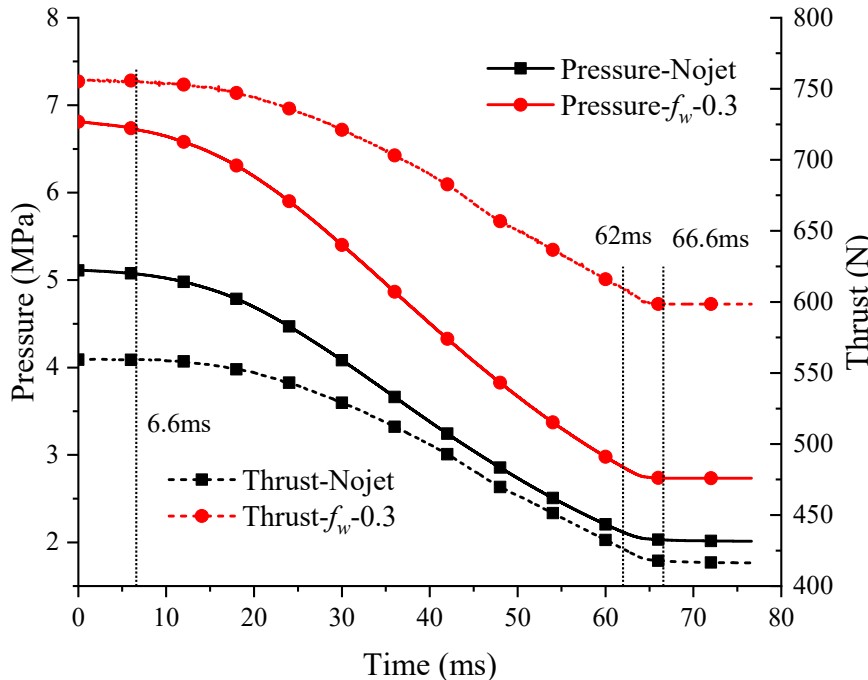

**Figure 19.** Variation of combustion chamber pressure and thrust during the backward movement process.

### 4.3. Effect of Injection Angle and Injection Port Position

The injection angle is a crucial parameter for injection mechanisms. Steady-state studies on both secondary injection and fluidic pintle systems suggest that reverse injection exhibits superior control effectiveness [14,18]. In this section, an analysis is conducted on the forward movement process of the pintle with secondary flow injection angles set at 60° (forward injection), 90° (vertical injection), and 120° (reverse injection), with a modified flow ratio $f_w = 0.3$. The pintle velocity and starting/stopping positions are consistent with those in the previous section. The results are presented in Figures 20 and 21.

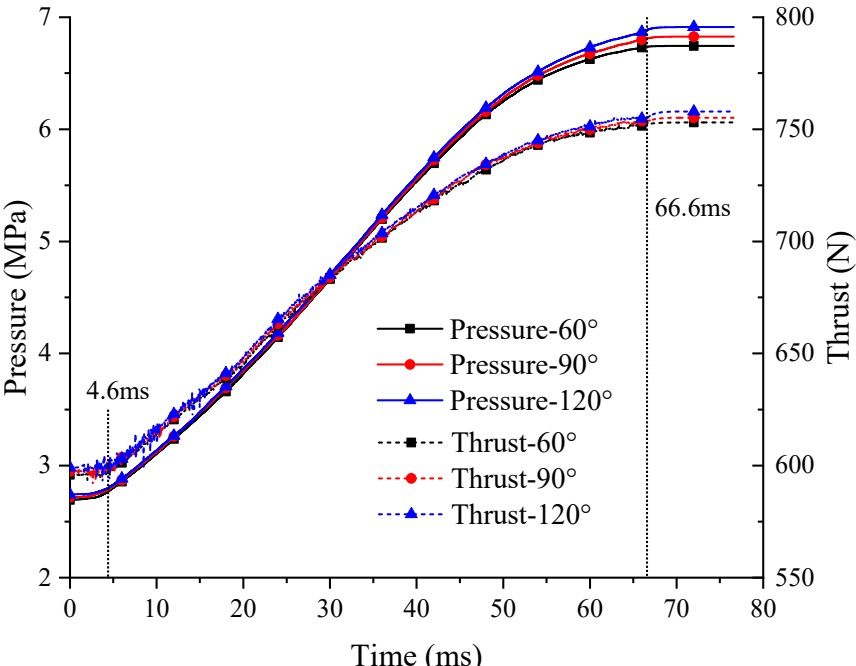

**Figure 20.** Combustion chamber pressure and thrust for different injection angles.

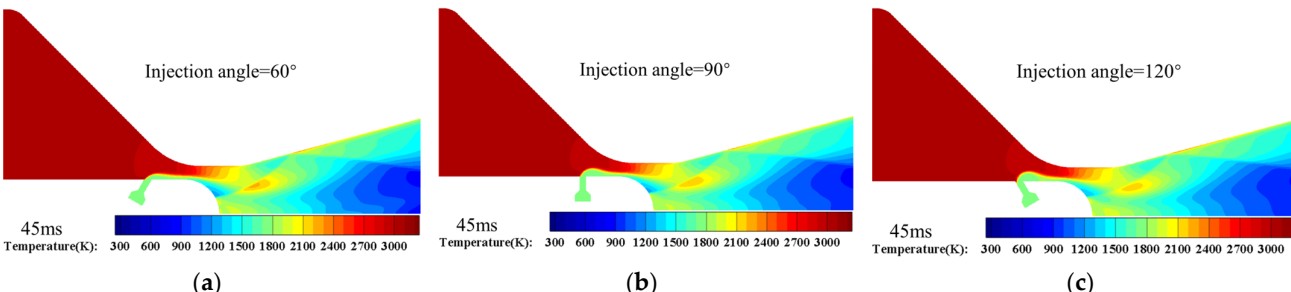

**Figure 21.** Temperature contours for different injection angles, t = 45 ms. (**a**) $\alpha = 60°$. (**b**) $\alpha = 90°$. (**c**) $\alpha = 120°$.

Figure 20 illustrates the variations in combustion chamber pressure and thrust for different injection angles. Although reverse injection has some advantage within the first 40 ms, it becomes more pronounced in the period from 40 to 66.6 ms. The results are generally consistent with static studies, showing that the effectiveness of reverse injection becomes more evident with smaller openings. However, the thrust control effectiveness of reverse injection is relatively moderate when considering the limited secondary flow rate.

The injection port position has been a focal point in the early-stage research, requiring considerations for both the constriction performance on the combustion gas and ensuring thermal protection of the pintle. In this section, an analysis is conducted on the forward movement process of the pintle with $L_i/R_p$ set at 0, 0.5, and 1. The injection angle is set at 90°, and the modified flow ratio is denoted as $f_w = 0.3$. The pintle velocity and starting/stopping positions are consistent with those in the previous section. The results are presented in Figures 22 and 23.

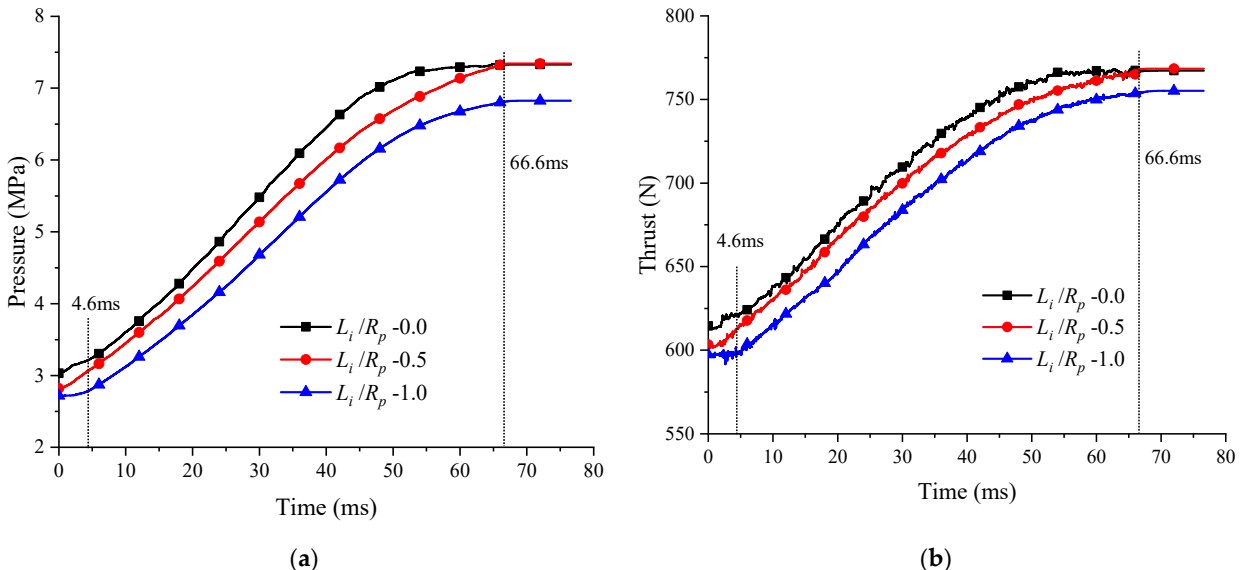

**Figure 22.** Variation in combustion chamber pressure and thrust with different injection port positions. (**a**) Combustion chamber pressure. (**b**) Thrust.

During the forward movement of the pintle, significant differences in pressure and thrust variations are observed for different injection port positions. In the initial state, a smaller $L_i/R_p$ (port closer to the pintle head) corresponds to a higher combustion chamber pressure and thrust. When $L_i/R_p = 0$, after 55 ms, there is a minimal increase in both pressure and thrust. This is attributed to the pintle entering the straight section of the nozzle throat, and as the pintle motion ceases, the geometry near the nozzle throat remains unchanged. Consequently, there is minimal alteration in the flow field, leading to a relatively stable combustion chamber pressure. For $L_i/R_p$ equal to 0.5 and 1, after the pintle

ceases its motion, the pressure and thrust gradually stabilize. This indicates that even when the pintle stops, the injection port remains at the upstream arc of the nozzle throat. At this point, the flow field near the throat is still not entirely stable, requiring some time after the pintle stops to achieve pressure stability. Comparatively, $L_i/R_p = 0.5$ achieves both the maximum thrust control range and exhibits good responsiveness.

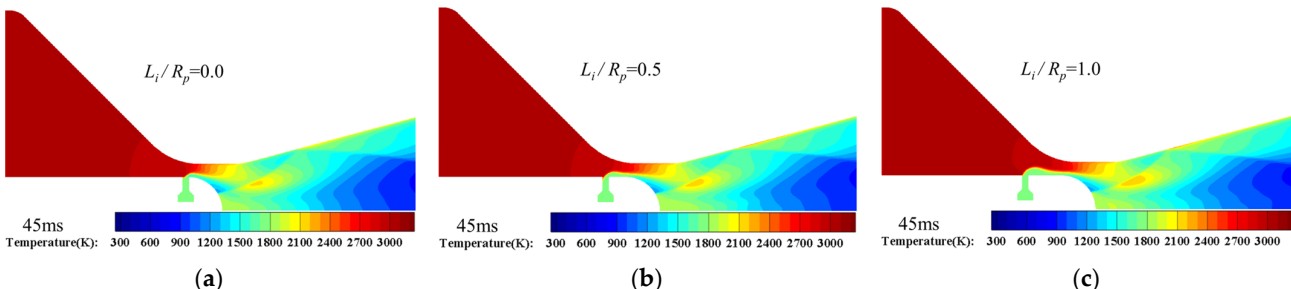

**Figure 23.** Temperature contours for different injection port positions at t = 45 ms. (**a**) $L_i/R_p = 0.0$. (**b**) $L_i/R_p = 0.5$. (**c**) $L_i/R_p = 1.0$.

Collectively, the location of the injection port exerts a noteworthy influence on both combustion chamber pressure and thrust. Proximity of the injection port to the pintle head augments the throttling efficacy on the primary flow, albeit at the expense of diminishing thermal protection for the pintle. Consequently, in the design phase, a holistic consideration encompassing both thermal protection and thrust control performance is imperative.

## 5. Conclusions

The fluidic pintle nozzle (FPN) is a newly proposed thrust control solution for solid rocket motors in recent years, offering advantages such as flexible control and thermal protection. This paper elucidates the thrust control principle of FPN and summarizes the wave structure inside FPN. Through transient numerical simulations using dynamic mesh, FPN is computationally analyzed to study its transient characteristics. The following conclusions are drawn from the study.

(1) The injection process in FPN can be roughly divided into three stages: the pressure propagation stage (combustion chamber pressure remains constant), pressure oscillation stage (combustion chamber pressure undergoes oscillations), and equilibrium stability stage (the combustion chamber pressure steadily rises), accounting for approximately 0.4%, 5.39%, and 94.21% of the total time, respectively.

(2) During the forward movement of the pintle, the combustion chamber pressure rapidly increases, with the rate of increase gradually decreasing (related to the upstream arc of the nozzle throat). Compared with the condition with maximum throat opening and no secondary flow, the thrust of the condition with minimum throat opening and 0.3-flow-ratio secondary flow is increased by 80.95%. In the backward movement of the pintle, the combustion chamber pressure gradually decreases, with the rate of decrease gradually increasing.

(3) Under the condition of a limited flow ratio, the injection angle of the secondary flow has little influence on the dynamic thrust control, but the control effect of reverse injection is more obvious when the throat opening is smaller. The closer the injection port is to the pintle head, the better the thrust control effect is, albeit at the cost of weakening the thermal protection of the low-temperature secondary flow.

**Author Contributions:** Conceptualization, D.Y.; methodology, D.Y. and Z.Z.; software, D.Y.; validation, Z.Z. and A.S.; formal analysis, D.Y. and F.L.; investigation, D.Y. and G.Z.; resources, D.Y. and L.Y.; data curation, D.Y. and S.M.; writing—original draft preparation, D.Y.; writing—review and editing, D.Y. and L.Y.; visualization, D.Y. and A.S. All authors have read and agreed to the published version of the manuscript.

**Funding:** This research was funded by the Fundamental Research Funds for the Central Universities (J2023-004) and the Fundamental Research Funds for the Central Universities (ZJ2023-013).

**Data Availability Statement:** The data presented in this study are available on request from the corresponding author.

**Conflicts of Interest:** The authors declare no conflicts of interest.

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
