# Peer review of "Transient Characteristics of Fluidic Pintle Nozzle in a Solid Rocket Motor"

_aerospace, doi:10.3390/aerospace11030243_

Round 1
Reviewer 1 Report
Comments and Suggestions for Authors
See attachment.

Reviewer 2 Report
Comments and Suggestions for Authors
In their article, the authors conduct a comprehensive review of Computational Fluid Dynamics (CFD) studies pertaining to fluidic pintle nozzles. They explore various aspects of the system through multiple CFD simulations, shedding light on its performance under different conditions. This thorough analysis contributes significantly to the understanding and advancement of fluidic pintle nozzle technology, providing valuable insights for its development and optimization.
There is a small error in Fig. 2, as it shows the radius of the combustion chamber as 60 mm, where it should be 30 mm.
The manuscript is very complete and shows lots of simulation results, with significant discussion of the system under different conditions.
Comments on the Quality of English LanguageA minor english review is recommended, as there are some small errors in the manuscript, i.e. "attitude" instead of "altitude" in line 37, or "contorl" instead of "control" in line 49.
Round 2
Reviewer 1 Report
Comments and Suggestions for Authors
All comments have been sufficiently adressed.